# Extensive age-dependent loss of antibody diversity in naturally short-lived turquoise killifish

**William John Bradshaw[1,2], Michael Poeschla[1], Aleksandra Placzek[1], Samuel Kean[1], Dario Riccardo Valenzano[1,2]***

[1]Max Planck Institute for Biology of Ageing, Cologne, Germany; [2]University of Cologne, Cologne, Germany

**Abstract** Aging individuals exhibit a pervasive decline in adaptive immune function, with important implications for health and lifespan. Previous studies have found a pervasive loss of immune-repertoire diversity in human peripheral blood during aging; however, little is known about repertoire aging in other immune compartments, or in species other than humans. Here, we perform the first study of immune-repertoire aging in an emerging model of vertebrate aging, the African turquoise killifish (*Nothobranchius furzeri*). Despite their extremely short lifespans, these killifish exhibit complex and individualized heavy-chain repertoires, with a generative process capable of producing millions of distinct productive sequences. Whole-body killifish repertoires decline rapidly in within-individual diversity with age, while between-individual variability increases. Large, expanded B-cell clones exhibit far greater diversity loss with age than small clones, suggesting important differences in how age affects different B-cell populations. The immune repertoires of isolated intestinal samples exhibit especially dramatic age-related diversity loss, related to an elevated prevalence of expanded clones. Lower intestinal repertoire diversity was also associated with transcriptomic signatures of reduced B-cell activity, supporting a functional role for diversity changes in killifish immunosenescence. Our results highlight important differences in systemic vs. organ-specific aging dynamics in the adaptive immune system.

## Editor's evaluation

This study introduces the killifish as a short-lived vertebrate model for immune aging and immunosenescence and characterizes the changes in the immune-repertoire during aging. This work provides an important first step in understanding how aging impacts the immune system in this model organism and will set the stage for many future studies.

## Introduction

The adaptive immune system undergoes a severe and systemic decline in proper function with age, resulting in higher susceptibility to a wide range of infections and decreased efficacy of vaccination in elderly individuals (*Ademokun et al., 2010*; *Kogut et al., 2012*; *Dunn-Walters and Ademokun, 2010*). In the humoral immune system, aging is accompanied by a decline in naïve B-cell output from the primary lymphoid organs; impaired production of specific antibodies in response to antigenic challenge; and a decline in antibody quality (*Ademokun et al., 2010*; *Kogut et al., 2012*; *Sasaki et al., 2011*; *Aberle et al., 2013*), as well as impairments in the establishment of novel immune memory (*Aberle et al., 2013*). These changes are major contributors to a generalized immunosenescent phenotype that significantly impairs health and quality of life in the elderly.

*For correspondence:
dvalenzano@age.mpg.de

The efficacy of the humoral immune system rests on its ability to generate an enormous array of different antibody sequences, with a correspondingly vast range of antigen specificities, and to progressively adjust the composition of this antibody population in response to antigen exposure (*Schatz and Swanson, 2011*; *Di Noia and Neuberger, 2007*; *Magor, 2015*; *Elhanati et al., 2015*). Sampling the resulting repertoire of antibody sequences in an individual using high-throughput sequencing can yield important insights into the diversity, clonal composition, and history of antibody-mediated immunity in that organism, as well as the effect of age, antigen exposure, and other factors on the diversity and functionality of the adaptive immune system (*Weinstein et al., 2009*; *de Bourcy et al., 2017*; *Miho et al., 2018*).

In humans, antibody-repertoire sequencing has uncovered a number of important age-related changes, including reduced numbers of clones and unique sequences, increased baseline mutation, more frequent and larger clonal expansions, impaired B-cell selection, and a shift toward the memory compartment (*de Bourcy et al., 2017*; *Jiang et al., 2013*; *Wang et al., 2014*). The responsiveness of the peripheral repertoire to vaccination is also impaired during aging (*de Bourcy et al., 2017*; *Wang et al., 2014*). While within-individual repertoire diversity declines with age, between-individual variability increases, with repertoires from older individuals differing more from one another than those from young individuals (*de Bourcy et al., 2017*).

Previous work in humans, however, has been limited by small sample sizes, a lack of temporal resolution, or a restriction to peripheral blood samples, which are known to systematically under-represent the majority of B-cells resident in other organs and tissues (*Siegrist and Aspinall, 2009*; *Tabibian-Keissar et al., 2016*). Very little is known about how repertoire aging differs between distinct organs; in particular, almost nothing is known about how aging affects antibody repertoires at mucosal surfaces, which represent a crucial interface between the body and its microbial environment (*Belkaid and Hand, 2014*; *Magadan et al., 2019*). Even less is known about how aging might affect the antibody repertoires of vertebrates other than mice and humans.

In this study, we introduce the naturally short-lived turquoise killifish (*Nothobranchius furzeri*) (*Bradshaw and Valenzano, 2020*; *Hu and Brunet, 2018*; *Poeschla and Valenzano, 2020*; *Reichwald et al., 2015*) as a model for adaptive immunosenescence. Here, we perform the first immune-repertoire sequencing experiments in this species, demonstrating that adult killifish express diverse and individualized heavy-chain repertoires that undergo rapid loss of diversity with age. The age-dependent loss of the antibody-repertoire diversity primarily affects the composition of expanded clones, with small naïve clones exhibiting much smaller age-related changes. By sequencing the repertoires of isolated intestinal samples, we further find that the killifish intestinal antibody repertoire exhibits much more dramatic age-dependent diversity loss than the body as a whole, possibly due to a much higher prevalence of expanded clones in the intestine, and that this loss of diversity is associated with gene expression changes indicating reduced B-cell activity. Taken together, our results reveal substantial differences between whole-body and organ-specific immune-repertoire aging, and establish the turquoise killifish as a powerful model for studying adaptive immune senescence.

## Results
### Establishing immunoglobulin sequencing in the turquoise killifish

To investigate the effect of age on the B-cell receptor repertoire diversity and composition in turquoise killifish, we implemented an RNA-based repertoire-sequencing protocol based on the published protocol of *Turchaninova et al., 2016*, using template switching (*Zajac et al., 2013*) to add unique molecular identifiers (UMIs) to each RNA transcript of the immunoglobulin heavy chain (*Figure 1*; *Bradshaw and Valenzano, 2020*) to correct for errors and biases in abundance arising during PCR and Illumina sequencing (*Vollmers et al., 2013*). To test the validity and replicability of results obtained using this protocol, we performed three replicate library preps on whole-body total RNA samples from four adult (8-week-old) adult male turquoise killifish from the short-lived GRZ strain (*Figure 1—figure supplement 1*). Independent repertoires from the same individual showed a high degree of similarity in their clonal composition, with an average inter-replicate correlation in clone size of $r = 0.89$ (*Figure 1—figure supplement 2*). Inter-repertoire divergences computed with the published repertoire dissimilarity index (RDI) metric (*Bolen et al., 2017*) consistently identified replicates from the same individual as much more similar than repertoires from different individuals

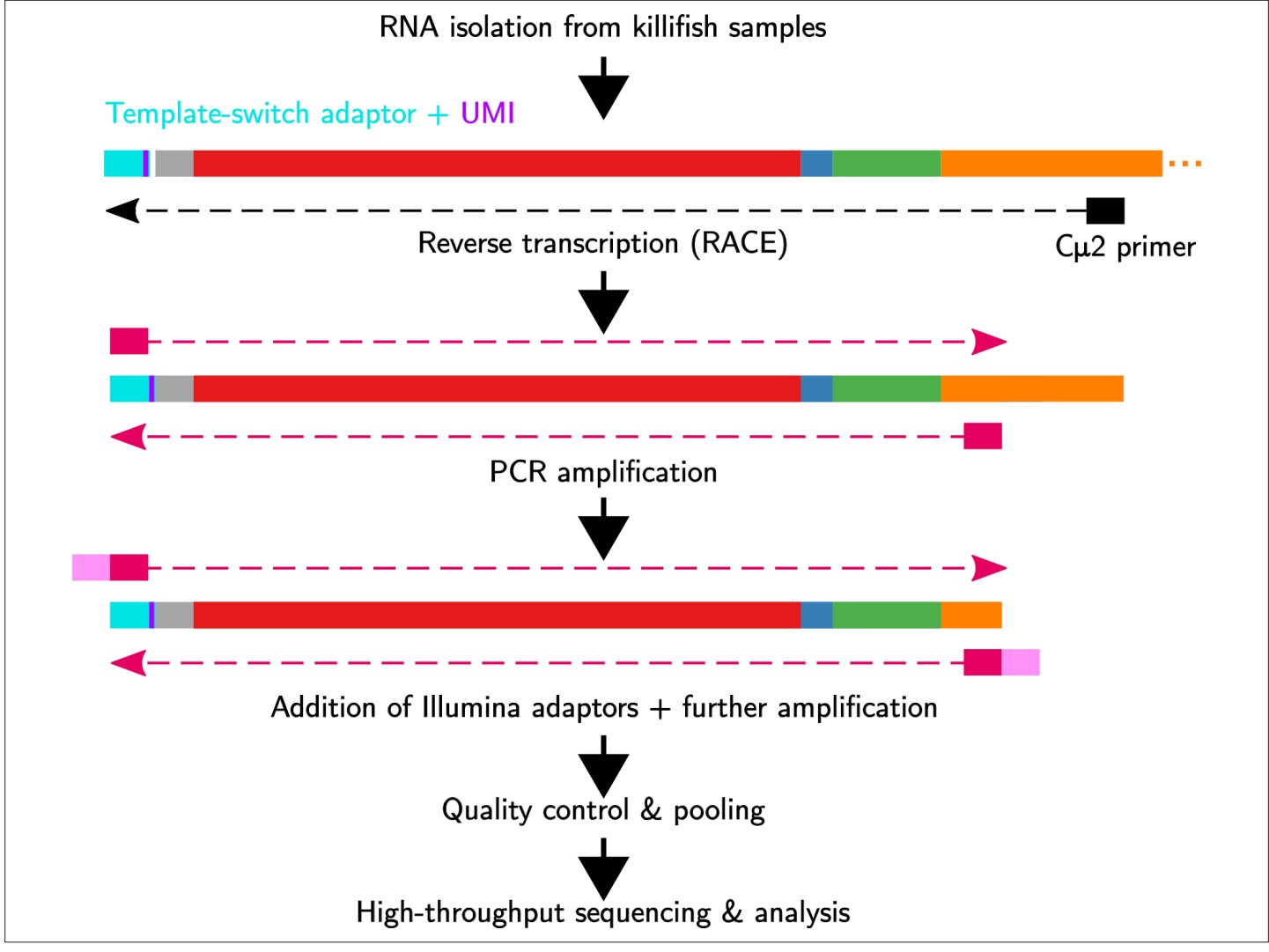

**Figure 1.** Immunoglobulin sequencing from turquoise-killifish total RNA samples. Each sample undergoes reverse transcription with template switching to attach a 5' adaptor sequence and unique molecular identifier (UMI), followed by multiple rounds of PCR amplification and addition of Illumina sequencing adaptors. Libraries are then pooled, undergo size selection, and are sequenced on an Illumina MiSeq sequencing machine.

The online version of this article includes the following figure supplement(s) for figure 1:

**Figure supplement 1.** Experimental design of the IgSeq pilot experiment in the turquoise killifish.

**Figure supplement 2.** Replicability of IgSeq results on total-body RNA.

**Figure supplement 3.** Clustering of replicates in control libraries.

(*Figure 1—figure supplement 3*), demonstrating that this protocol is capable of accurately and reproducibly reconstructing the expressed heavy-chain repertoires of individual killifish.

## Aging in whole-body killifish repertoires

To investigate the effect of age on the structure and diversity of killifish antibody repertoires, we performed whole-body immunoglobulin sequencing on 32 adult male turquoise killifish from the short-lived GRZ strain (*Hu and Brunet, 2018*) at four different ages from early adulthood to late life (*Figure 2A* and *Supplementary file 2a-d*). The repertoire of each individual comprised some number of unique heavy-chain sequences, each of which could be classified by clonal identity (the 'clonal repertoire') and V/J usage (the 'VJ repertoire').

The diversity of a population is a measure of the number (a.k.a. the *richness*) and relative frequency of different subdivisions within that population. For B-cell repertoires, diversity can be calculated over

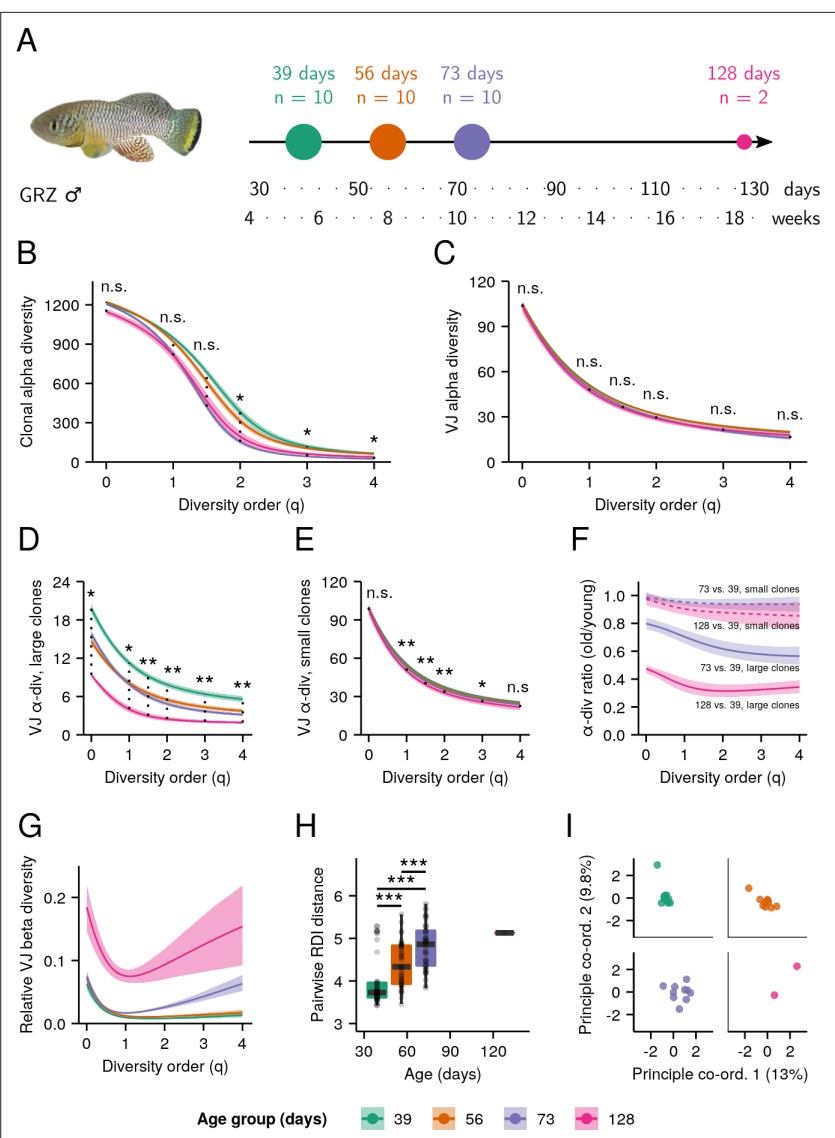

**Figure 2.** Aging in whole-body killifish *IGH* repertoires. (**A**) Experimental design. Adult male GRZ-strain turquoise killifish were sacrificed at 39, 56, 73, and 128 days post-hatching, flash-frozen and homogenized. (**B–E**) Alpha-diversity spectra, indicating average within-individual repertoire diversity for each age group and diversity order (*: $0.05 \leq 0.01$, **: $0.01 \leq p \leq 0.001$, Kruskal-Wallis permutation test, Appendix 1—note 7). (**B**) Clonal alpha-diversity spectra. (**C**) VJ alpha-diversity spectra, all clones. (**D**) VJ alpha-diversity spectra, large clones (>4 unique sequences) only. (**E**) VJ alpha-diversity spectra, small clones (<5 unique sequences) only. (**F**) VJ alpha-diversity ratios for old vs. young killifish at each diversity order, for small (dashed lines) or large (solid lines) clones. Color indicates the older age group being compared to young (39 days) fish. (**G**) Normalized VJ beta-diversity spectra, indicating between-individual variability in repertoire composition for each age group and diversity order. (**H**) Distributions of pairwise repertoire dissimilarity index (RDI) distances between individuals in each age group (***: $p \leq 0.001$, Mann-Whitney U tests for pairwise age differences), based on the VJ composition of each individual's repertoire. (**I**) Principal coordinate analysis (PCoA) of pairwise RDI distances for each age group, visualizing the progressively greater dispersion seen at later ages. Each curve in (**B–G**) represents the mean across 2000 bootstrap replicates (Appendix 1—note 7); shaded regions indicate 95% confidence intervals over the same.

The online version of this article includes the following figure supplement(s) for figure 2:

**Figure supplement 1.** Clonal and V/J diversity in antibody repertoires.

**Figure supplement 2.** Individual diversity spectra in killifish whole-body repertoires.

**Figure supplement 3.** p-Values of Kruskal-Wallis permutation tests for an age effect (Appendix 1—note 7) in whole-body killifish *IGH* repertoires at different Hill diversity orders.

**Figure supplement 4.** Clone size distributions in killifish whole-body repertoires.

the different clonal lineages detected in a sample, or over different variable-region gene segment combinations.

To quantify immune-repertoire diversity in killifish heavy-chain repertoires, we computed Hill diversity spectra (Appendix 1—note 1), which provide a holistic overview of the diversity structure of a population (in this case, a repertoire) (*Miho et al., 2018*; *Hill, 1973*; *Jost, 2006*; *Jost, 2007*). Briefly, each Hill spectrum reports the 'effective richness' of a repertoire across a range of *diversity orders*; higher effective richness corresponds to a more diverse repertoire. At low diversity orders, species of different sizes are weighted more equally in the diversity calculation, while at higher orders, less-abundant species are progressively downweighted relative to more-abundant groups (Appendix 1—note 1). For the clonal repertoire, each clonotype (set of unique sequences descended from a single naïve ancestor B-cell) was designated a separate species; for the VJ repertoire, clonotypes with the same V/J identity were grouped together.

Separate spectra were computed for the clonal and VJ repertoires of each individual, and these individual spectra were used to compute averaged alpha-diversity spectra for each age group and repertoire type. For each diversity order, we tested for a significant age effect on repertoire diversity using a permutation test on the Kruskal-Wallis *H* statistic (Materials and methods).

The clonal diversity (*Figure 2—figure supplement 1*) of the whole-body killifish repertoire exhibited a significant decline with age (p < 0.05) at high diversity orders (*Figure 2B*, *Figure 2—figure supplements 2–3*), indicating a significant and extremely rapid age-related decline in the diversity of the largest B-cell clones. In contrast, lower-order clonal diversity exhibited no significant change with age, suggesting that the *overall* composition of the whole-body repertoire remains relatively unchanged. Since the B-cell clonal repertoire is overwhelmingly dominated by small, predominantly naïve clones (*Figure 2—figure supplement 4A*), low-order clonal diversity measurements are primarily driven by changes in the diversity of small clones. As such, these results indicate that the composition of small clones in the killifish antibody repertoire is much less sensitive to the effects of aging than that of large, expanded clones.

In contrast with the rapid age-related declines observed in high-order clonal diversity, the VJ diversity of the killifish repertoire exhibited no significant age-related change at any diversity order (*Figure 2C*). Examining the clone-size distribution of each V/J combination (*Figure 2—figure supplement 4B*) revealed that even the largest V/J combinations in each age group are overwhelmingly dominated by small clones, suggesting that the observed lack of an age effect on VJ diversity was due to the observed age-insensitivity of small clones (*Figure 2B*). To test this hypothesis, we filtered the repertoire dataset to separate sequences from small and large clones and computed VJ diversity repertoires for each subset (*Figure 2D–E*). While both small and large clones in isolation showed a significant age effect on VJ diversity, the relative reduction in VJ alpha diversity with age was dramatically stronger for large clones, an effect observed across all diversity orders (*Figure 2F*). As suggested by the clonal-diversity results, therefore, the repertoire diversity of large (expanded) clones in the killifish whole-body repertoire appears to be far more age-sensitive than that of small (predominantly naïve) clones.

In addition to the average within-individual diversity of a population (alpha diversity), the between-individual variation in composition (beta diversity) can provide important insights into repertoire development and evolution. Previous studies of human peripheral blood repertoires have suggested a decrease in alpha diversity but an *increase* in beta diversity with age (*Gibson et al., 2009*; *de Bourcy et al., 2017*). In our dataset, VJ beta-diversity spectra (Appendix 1) indicated a large age-related increase in beta diversity across a wide range of diversity orders (*Figure 2G*), indicating a similar pattern of progressive individualization in repertoire composition with age. Concordantly, older killifish also exhibited significantly greater pairwise RDI distances (*Bolen et al., 2017*), indicating progressive divergence in repertoire composition (*Figure 2H–I*). As in humans, therefore, younger killifish exhibit antibody repertoires that are significantly more similar to one another, which then become increasingly distinct and individualized as the cohort increases in age.

## The killifish generative repertoire

The naïve sequence diversity of the antibody heavy-chain repertoire depends on the molecular processes underlying the generation of novel sequences in developing B-cells: random selection of V, D, and J segments during VDJ recombination; deletions and palindromic (P-) insertions at the ends

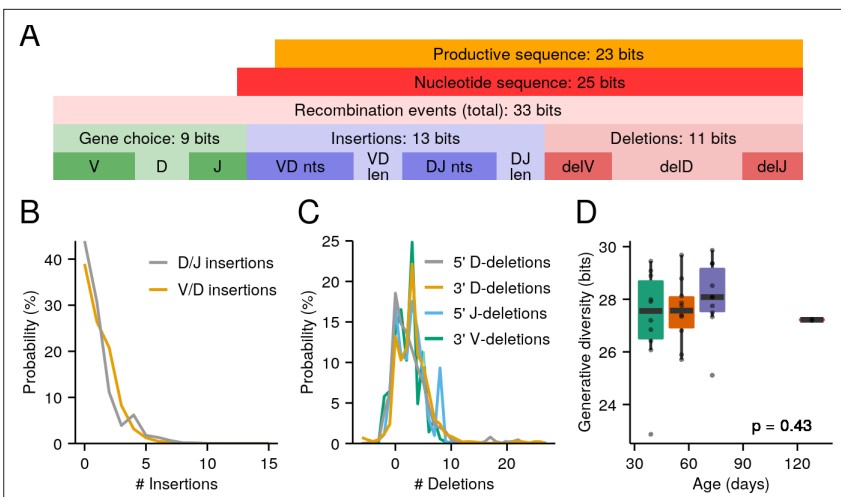

**Figure 3.** The killifish generative repertoire. (**A**) Entropy composition of the generative process from four 8-week-old GRZ-strain adult male turquoise killifish. (**B**) Probability distributions of junctional N-insertions in the same dataset. (**C**) P-insertions and deletion distributions inferred from the same dataset, with P-insertions modeled as negative deletions. (**D**) Boxplots of total recombination entropy values for models inferred separately for each individual in the 32-individual aging cohort (p = 0.43, Kruskal-Wallis one-way analysis of variance [ANOVA] for an age effect).

The online version of this article includes the following figure supplement(s) for figure 3:

**Figure supplement 1.** Individual IGoR-inferred insertion/deletion distributions (*Figure 3A*) for individuals in the pilot dataset (*Figure 1—figure supplement 1*).

**Figure supplement 2.** Individual IGoR-inferred segment-usage distributions (*Figure 3B*) for individuals in the pilot dataset (*Figure 1—figure supplement 1*).

**Figure supplement 3.** Individual IGoR-inferred insertion/deletion distributions (*Figure 3A*) for all individuals in the whole-body killifish dataset (*Figure 2A* and *Supplementary file 2c*).

**Figure supplement 4.** Individual IGoR-inferred segment-usage distributions (*Figure 3B*) for all individuals in the whole-body killifish dataset (*Figure 2A* and *Supplementary file 2c*).

**Figure supplement 5.** Boxplots of the entropy contributions of (**A**) gene choice, (**B**) N-insertions, and (**C**) P-insertions and deletions to the generative repertoires of individual turquoise killifish at different ages (*Figure 2A* and *Supplementary file 2c*).

of conjoined segments; and nonpalindromic (N-) insertions between segments (*Schatz and Swanson, 2011*; *Schroeder and Cavacini, 2010*). Each of these contributes diversity to the overall generative process, increasing the variety of sequences that can be generated. Excluding nonfunctional sequences, the human generative process has an estimated Shannon entropy of roughly 70 bits, corresponding to a first-order Hill diversity of roughly $10^{21}$ possible unique sequences (*Elhanati et al., 2015*). However, little is known about how this generative diversity varies across species, or how it changes during aging.

To gain insight into these generative processes in the turquoise killifish, we used IGoR (*Marcou et al., 2018*) to infer models of sequence generation from killifish repertoire data. In training these models, we restricted the dataset to nonfunctional naïve sequences, in order to avoid distortions introduced by positive and negative selection in the primary lymphoid organs (*Elhanati et al., 2015*; *Marcou et al., 2018*). As is often the case with RNA data, the number of naïve nonfunctional sequences available per individual was frequently low; hence, to better capture low-probability events in the generative process, we inferred models from pooled data from multiple individuals in the same age group. As the parameters of the generative model are typically very similar across conspecific individuals (*Figure 3—figure supplements 1–2*), especially in an inbred line, pooling data like this is a useful way to infer more robust generative models using IGoR (Marcou 2019, personal communication).

To model the generative process in its baseline state, we first inferred a model of the killifish generative repertoire from the four 8-week-old adult male individuals used in the pilot study (*Figure 1—figure supplement 1*). Using this model, we estimated a total raw entropy for the killifish generative

repertoire of roughly 33 bits (*Figure 3A*). Of these 33 bits, roughly 8 arise from variability in VDJ segment choice, 12 from variability in the number and composition of junctional N-insertions, and 11 from P-insertions and deletions. Accounting for convergent production of identical sequences via different recombination events, and for events that give rise to nonfunctional nucleotide sequences (e.g. due to frame shift) reduced this initial raw estimate by 10 bits.

Before initial selection in the primary lymphoid organs, therefore, the killifish generative process has an estimated Shannon entropy of roughly 23 bits (*Figure 3A*), corresponding to a first-order Hill diversity of roughly $10^7$ possible unique sequences. While, as in humans, only a small fraction of potential diversity will actually be generated in any single individual, this nevertheless represents a highly complex and sophisticated system capable of generating highly individualized *IGH* repertoires.

While impressive, the potential generative diversity of the killifish repertoire is nevertheless vastly lower than in humans, with a difference in productive generative entropy of almost 50 bits (*Elhanati et al., 2015*). While all components of the generative process exhibit lower entropies in killifish than in humans, by far the greatest difference lies in the junctional N-insertions, which contribute almost 40 bits more to the generative entropy of the human repertoire than that of killifish. The difference in the productive generative entropy between killifish and human arises from the distributions of N-insertions inferred from killifish and human data: in humans, these distributions peak at around 5 nt per junction and often yield insertions of 10–20 nt (*Elhanati et al., 2015*), while in killifish the insertion distribution peaks at 0 nt per junction, and sequences with more than 5 nt of insertions at either junction are very rare (*Figure 3B* and *Figure 3—figure supplement 1*). Since N-insertions are the dominant source of sequence diversity in human repertoires, the large reduction in N-insertions in killifish relative to humans unsurprisingly results in a much lower overall generative diversity for the killifish adaptive immune system.

The relative lack of change in the small-clone antibody repertoire in older turquoise killifish (*Figure 2B–F*) suggested to us that the diversity of the generative process in the primary lymphoid organs might remain relatively intact throughout the killifish lifespan. To test this hypothesis, we trained separate IGoR models for each individual in the 32-fish aging cohort (*Figure 2A*, *Figure 3—figure supplements 3–4*) and tested for an effect of age on the generative diversity inferred for each individual. As expected, no age effect was found in either total generative diversity (*Figure 3D*) or the contributions of different diversification processes (*Figure 3—figure supplement 5*). It therefore appears that, while some aspects of the killifish antibody repertoire certainly decline with age, the entropy of the generative process is not among them.

## Effect of age and microbiota transfer on killifish intestinal repertoires

The populations of B-lymphocytes associated with mucosal epithelia play a crucial role in defending the body from pathogenic threats (*Magadan et al., 2019*), as well as in regulating the composition of resident microbial populations (*Belkaid and Hand, 2014*). Despite the importance of these distinctive B-cell compartments, relatively little is known about the structure of their antibody repertoires (*Magadan et al., 2019*), and still less about how these repertoires change with age.

As the site of the greatest microbiota diversity, the intestine is of particular relevance as an important and distinctive immune environment. Previous work on the killifish gut microbiota (*Smith et al., 2017*) has shown that it declines in alpha diversity and increases in beta diversity with age, patterns that mirror the changes seen in the whole-body composition of the killifish antibody repertoire (*Figure 2*). Transfer of intestinal content from young to middle-aged fish has also been shown to extend lifespan (*Smith et al., 2017*). Given these findings, and the intimate relationship between intestinal lymphocytes and gut bacteria (*Belkaid and Hand, 2014*), we investigated the effect of aging and microbiota transfer on the immune repertoires of gut-resident B-cell populations.

Using intestinal total RNA isolated by *Smith et al., 2017*, we sequenced the intestinal *IGH* repertoires of eighteen male GRZ-strain individuals, including four untreated 6-week-old individuals and fourteen 16-week-old individuals from various microbiota-transfer treatment groups (*Supplementary file 2b*), and investigated the effect of age and treatment condition on repertoire diversity in the killifish intestine.

Contrary to our expectations, neither the alpha nor beta diversity of the killifish intestinal repertoire were significantly affected by microbiota transfer, with no significant difference in clonal diversity, VJ diversity, or RDI distance measures (*Figure 4—figure supplements 1–3*). In sharp contrast to the

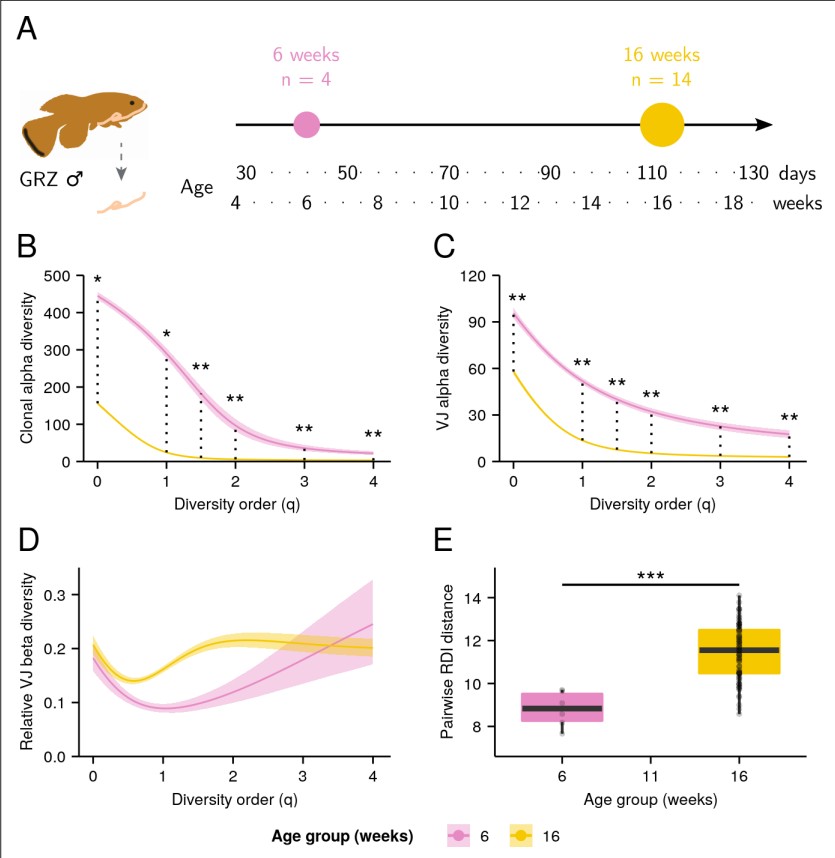

**Figure 4.** Aging in killifish intestinal repertoires. (**A**) Experimental design. Adult male GRZ-strain turquoise killifish were sacrificed at 6 and 16 weeks' post-hatching, and total RNA was extracted from the dissected intestine. (**B–C**) Alpha-diversity spectra, indicating average within-individual repertoire diversity for each age group and diversity order (*: $0.05 \leq 0.01$, **: $0.01 \leq p \leq 0.001$, Kruskal-Wallis permutation test, Appendix 1—note 7). (**A**) Clonal alpha-diversity spectra. (**B**) VJ alpha-diversity spectra, all clones. (**D**) Normalized VJ beta-diversity spectra, indicating between-individual variability in repertoire composition for each age group and diversity order. (**E**) Distribution of pairwise repertoire dissimilarity index (RDI) distances between killifish intestinal repertoires at different ages (***: $p \leq 0.001$, Mann-Whitney U tests for pairwise age differences). Each curve in (**A–C**) represents the mean across 2000 bootstrap replicates (Appendix 1—note 7); shaded regions indicate 95% confidence intervals over the same.

The online version of this article includes the following figure supplement(s) for figure 4:

**Figure supplement 1.** Effect of microbiota transfer on the intestinal repertoires of 16-week-old turquoise killifish.

**Figure supplement 2.** Individual diversity spectra in killifish intestinal repertoires.

**Figure supplement 3.** p-Values of Kruskal-Wallis permutation tests (Appendix 1—note 7) for (*left*) an effect of microbiota transfer treatment group or (**B**) an age effect in intestinal killifish *IGH* repertoires at different Hill diversity orders.

**Figure supplement 4.** Principal coordinate analysis (PCoA) of pairwise repertoire dissimilarity index (RDI) distances for each age group in the killifish intestinal dataset (***Figure 4E***, ***Supplementary file 2b***).

whole-body data, however, there was a strong and significant decline in both clonal and VJ alpha diversity with age across all diversity orders (***Figure 4B–C***, ***Figure 4—figure supplements 2–3***), even without partitioning by clone size. This age-related decline in alpha diversity was consistently far more dramatic than that observed in the whole-body samples at any diversity order. The B-cells of the killifish intestine, therefore, exhibit a much stronger age-dependent decline in repertoire diversity than is seen in the killifish body overall.

While results from beta-diversity spectra showed large increases in beta diversity with age at some diversity orders but not at others (***Figure 4D***), the median pairwise RDI distance between individual gut repertoires increased substantially and significantly with age (***Figure 4E*** and ***Figure 4—figure***

*supplement 4*), suggesting that, as in the whole body, killifish intestinal repertoires become increasingly distinct and individualized as they age.

One potential explanation for the stronger age-related drop in alpha diversity of intestinal samples is as a consequence of the constant strong antigen exposure experienced by intestinal B-cells, as a result of their interaction with the gut microbiota. This exposure could drive high levels of antigen-dependent clonal expansion, resulting in a greater loss in repertoire diversity (*Caruso et al., 2009*). Another explanation, not mutually exclusive with the first, is that the gut has different clone-size distribution relative to the whole body. Unlike the whole-body repertoire, the gut does not include the primary lymphoid organs, and so would be expected to be far less dominated by small, naïve clones. Since the population of large clones appears to be more prone to reductions in diversity with age than that of small clones (*Figure 2*), the stronger overall age-related diversity loss found in the gut repertoire could be a consequence of this greater relative prevalence of large clones.

Rarefaction analysis of clonal counts in whole-body and intestinal repertoires showed that the latter indeed contained far fewer small clones, resulting in a much higher proportion of large clones (*Figure 5A*). If this difference in clonal composition, rather than some functional difference between intestinal and other B-cells, is primarily responsible for the apparent difference in aging phenotypes between whole-body and intestinal repertoires, we would expect to find a faster rate of clonal diversity loss during aging in intestinal repertoires at low diversity orders (which are dominated by small clones in whole-body samples), but not at high orders (which are dominated by large clones in both sample types). Similarly, we would expect to find faster loss in intestinal samples of V/J diversity considered over all clones, but not when the V/J diversity calculation is restricted to large clones alone.

To test these hypotheses, we normalized the diversity measurements from each dataset by the mean diversity of the youngest group in that dataset, then fit generalized linear models for different diversity orders and methods of measuring diversity (*Figure 5B* and *Figure 5—figure supplement 1*), testing for a significant interaction between sample type (i.e. gut vs. whole body) and the effect of age on repertoire diversity. Gut samples exhibited significantly higher rates of age-dependent diversity loss under low-order clonal-diversity or total VJ-diversity measures, that is, those metrics for which clones of all sizes were included in the diversity calculation. Conversely, there was no significant difference in rate of diversity loss between sample types for higher-order clonal-diversity measures, nor for V/J-diversity measures restricted to only large clones, indicating that large clones undergo similar rates of age-dependent diversity loss in both sample types. These results closely match the predictions of the clonal-composition model: large clones in both gut and whole-body samples exhibit similarly strong aging phenotypes, but the higher proportions of large clones in gut samples result in these strong phenotypes manifesting more strongly in the behavior of the repertoire as a whole. It therefore appears that, as in whole-body samples, age-dependent diversity loss in killifish intestinal repertoires is primarily a phenomenon of mature, expanded clones.

## Functional correlates of repertoire diversity in killifish

Early work in killifish identified a number of age-associated phenotypes suggestive of immune decline, including thymic degeneration and increased incidence of lymphoma (*Cooper et al., 1983*). More recently, comparison of young vs. old killifish intestines found a marked age-related increase in the pathogenicity of the killifish gut microbiome, alongside an increase in expression of inflammatory markers, suggesting a decline in the intestinal immune system's ability to maintain a healthy microbial community (*Smith et al., 2017*).

These results suggest that the turquoise killifish undergoes rapid functional immune decline with age. Since repertoire diversity also declines with age, this indicates that markers of functional immune decline are coincident with a reduction in repertoire diversity. These results are consistent with similar findings in humans (*de Bourcy et al., 2017*). However, in the absence of repertoire diversity data, these results do not necessarily imply a direct association between diversity and immune function in aging killifish.

To investigate the relationship between repertoire diversity and immune function more closely, we utilized previously collected intestinal RNA-seq data from the same cohort of killifish used in our intestinal antibody-repertoire analysis (*Smith et al., 2017*). Using these data alongside our repertoire diversity calculations, we carried out a differential expression analysis of transcript abundance with respect to repertoire diversity for six different diversity orders, controlling for age (Materials and

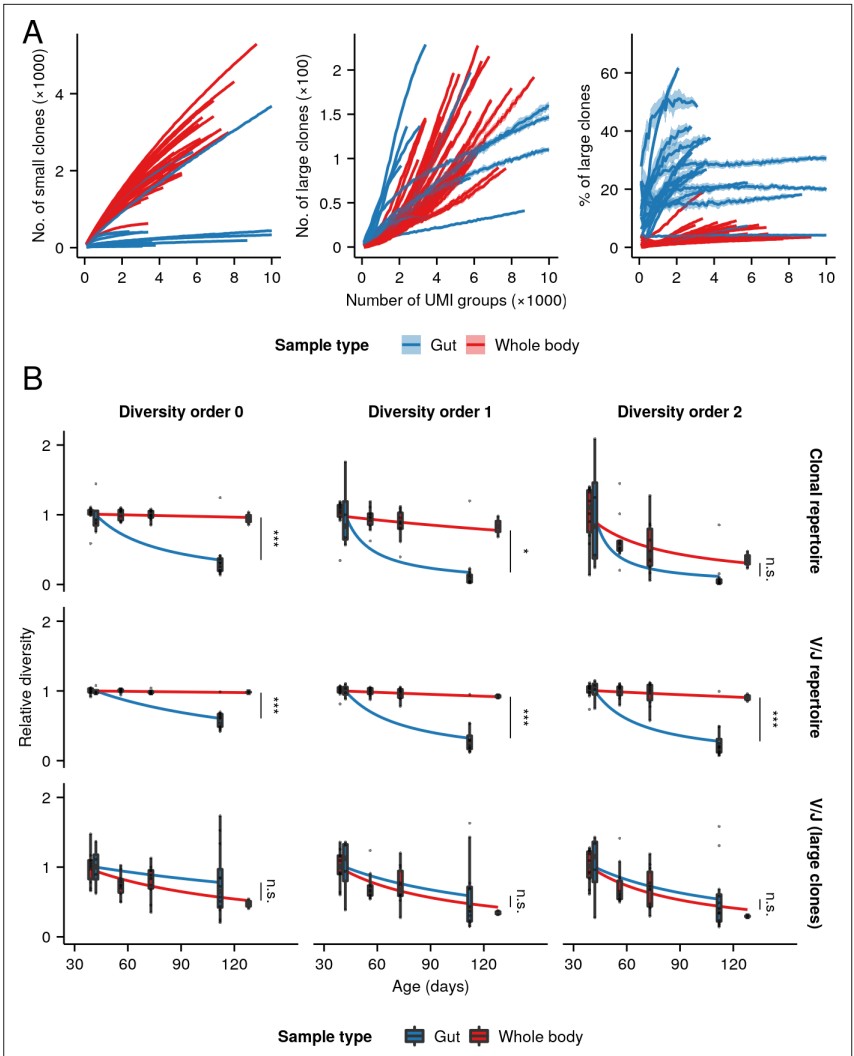

**Figure 5.** Relative clonal composition and aging phenotypes of whole-body and intestinal repertoires. (**A**) Rarefaction analysis of clonal composition of antibody repertoires from whole-body and intestinal samples, showing the average number of small (left, <5 unique sequences) and large (middle, 5 unique sequences) clones for each individual across 20 independent replicates at each sample size, as well as the average proportion of all clones in each repertoire which are large (right). Shaded regions around each line show the region within one standard deviation of the mean value. (**B**) Boxplots of individual diversity measurements of repertoires from each age group in the whole-body and intestinal datasets, divided by the mean diversity of the youngest age group in each dataset. Fitted curves show the maximum-likelihood prediction of a gamma-distributed generalized linear model of diversity vs. age and sample type for the whole-body and intestinal dataset, relative to the average diversity of the youngest age group in each experiment, testing for a significant effect of sample type on the rate of diversity change with age (Student's t-test,*: $0.01< p\ 0.05$; ***: $p\ 0.001$).

The online version of this article includes the following figure supplement(s) for figure 5:

**Figure supplement 1.** Extended boxplots of individual diversity measurements of repertoires from each age group in the whole-body and intestinal datasets, divided in each case by the mean diversity of the youngest age group in that dataset.

methods). We then performed gene set enrichment analysis (GSEA) to identify gene ontology (GO) terms associated with higher or lower repertoire diversity, across a variety of diversity orders.

The GSEA identified a number of GO terms related to immune function that were significantly associated with increased repertoire diversity (*Figure 6*, *Figure 6—figure supplements 1–3*). Most strikingly, 'B-cell receptor signaling pathway' was the most strongly enriched term for all six diversity orders analyzed, often by a substantial margin. 'B-cell proliferation' was also consistently highly

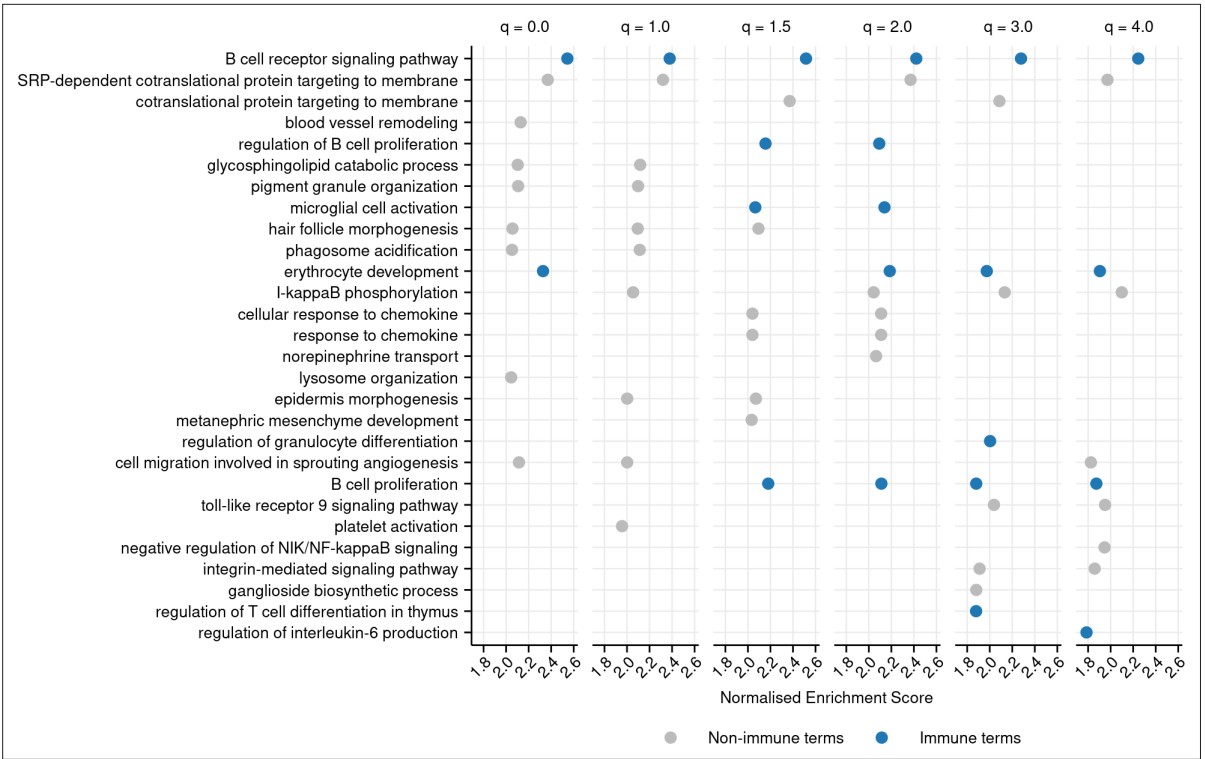

**Figure 6.** Top 10 most positively enriched gene ontology (GO) terms associated with each diversity order in turquoise killifish, controlling for age, ranked by normalized enrichment score in descending order (Materials and methods). Immune terms are highlighted in blue. Terms that are significantly positively enriched for a given diversity order, but not in the top 10, are not shown, even if they fall in the top 10 terms for other orders.

The online version of this article includes the following figure supplement(s) for figure 6:

**Figure supplement 1.** Top 10 most negatively enriched gene ontology (GO) terms associated with each diversity order in turquoise killifish, controlling for age, ranked by normalized enrichment score in ascending order (Materials and methods).

**Figure supplement 2.** Plot of all significantly positively enriched gene ontology (GO) terms (FDR-adjusted p-value ≤ 0.05) associated with each diversity order in turquoise killifish, controlling for age, ranked by normalized enrichment score in descending order (Materials and methods).

**Figure supplement 3.** Plot of all significantly negatively enriched gene ontology (GO) terms (FDR-adjusted p-value ≤ 0.05) associated with each diversity order in turquoise killifish, controlling for age, ranked by normalized enrichment score in ascending order (Materials and methods).

enriched, showing significant positive enrichment for five diversity orders (all except 1.0) and falling in the top 10 most positively enriched terms for four (*Figure 6*). Other immune terms that were significantly positively associated with repertoire diversity across at least four diversity orders include 'leukocyte migration', 'lymphocyte activation', 'leukocyte differentiation', and 'regulation of interleukin-6 production' (*Supplementary file 3a*). A decline in repertoire diversity is thus associated with a decline in B-cell immune activity in killifish intestine, supporting a functional role for diversity changes in killifish immunosenescence.

## Discussion

The turquoise killifish is the shortest-lived vertebrate that can be bred in captivity (*Cellerino et al., 2016*; *Harel and Brunet, 2015*), with a median lifespan in the short-lived GRZ strain of about 4 months. Despite this, our findings show that the life of a turquoise killifish provides ample time both to develop a complex, diverse, and individualized IgM heavy-chain repertoire (*Figure 3A*, *Figure 1— figure supplements 2–3*), and for that repertoire to decline significantly in diversity with age.

These age-associated diversity changes appear to be driven primarily by expanded, antigen-experienced clones, with little observed change in either the diversity of small naïve clones or the entropy of the heavy-chain sequence generation process. This lack of change in small-clone diversity, however, does not necessarily imply that B-cell development is unchanged in aging killifish: it is possible, for example, that a decline in efficiency of B-cell output from primary lymphoid organs

is offset by the continuous growth in body size observed throughout the killifish lifespan. Further research into killifish lymphopoiesis will shed light on the relationship between age and the naïve B-cell repertoire.

As early as 2009, *Caruso et al., 2009*, hypothesized that the mucosal adaptive immune system might exhibit particularly strong loss of diversity with age. In what is, to our knowledge, the first published test of this hypothesis, we sequenced the heavy-chain repertoires of isolated killifish gut samples, finding that they do indeed exhibit particularly strong diversity changes with age. However, this difference between the gut and whole-body repertoires appears to result from a difference in clonal composition, rather than in the behavior of any particular clonal subset, suggesting this difference may have less to do with the specifics of the mucosal environment than the location of the primary lymphoid organs. Whatever its source, this age-dependent loss of mucosal repertoire diversity could have important consequences for the gut's capacity to respond to novel antigens. Future investigation of immune-repertoire aging in a wider variety of mucosal and non-mucosal organs will help disentangle the effects of spatial context on adaptive immunosenescence and provide a clearer picture of the impact of mucosal microbiota.

While our results demonstrate that killifish repertoire diversity declines rapidly with age, the effects on immune function are less clear. Gene expression data from our intestinal cohort indicates that greater intestinal repertoire diversity is associated with gene expression changes indicating greater B-cell receptor signaling activity, lymphocyte activation, and defense responses, suggesting that the decline in diversity seen with age is associated with a decline in immune function. While not necessarily causal, these associations support the biological relevance of repertoire diversity as a metric of immune function. Nevertheless, future experiments should directly investigate the causal association between repertoire diversity and immune function in killifish.

Apart from the nervous system itself, no other system in the vertebrate body exhibits such complex learning and memory behavior as the adaptive immune system. The age-related decline in the functionality of this system is a major cause of mortality and morbidity in the elderly. Our results firmly establish the value of the turquoise killifish as a model for investigating this important and complex process, and demonstrate the importance of studying immune aging in compartments other than peripheral blood. Future experiments in this system have the potential to greatly expand our knowledge of the mechanisms, spatial distribution, and temporal progression of immune-repertoire aging, with potentially vital implications for the future treatment of immunosenescent phenotypes.

## Materials and methods
### Fish husbandry and sample preparation
Male turquoise killifish (*N. furzeri*, GRZ-AD strain) from a single hatching cohort were raised under standard husbandry conditions (*Dodzian et al., 2018*) and housed from 4 weeks' post-hatching in individual 2.8 l tanks connected to a water-recirculation system. Fish received 12 hr of light per day on a regular light/dark cycle, and were fed bloodworm larvae and brine shrimp nauplii twice a day during the week and once a day during the weekend (*Smith et al., 2017*; *Dodzian et al., 2018*).

After being sacrificed in 1.5 g/l tricaine solution at room temperature tank water (*Carter et al., 2010*), fish (*Supplementary file 2c*) were flash-frozen in liquid nitrogen and ground to a homogenous powder with a pestle in a liquid-nitrogen-filled mortar. The powder was mixed thoroughly and stored at –80°C prior to RNA isolation. Intestinal total RNA for the gut experiments was provided by *Smith et al., 2017*.

### Immunoglobulin sequencing
Total RNA from whole-body killifish samples was isolated using QIAzol lysis reagent (QIAGEN, 1 ml of reagent per 0.1 g of homogenized tissue) and isopropanol precipitation; gut RNA from microbiota-transfer experiments (*Smith et al., 2017*) was already prepared and available. Quantification of RNA samples was performed with the Qubit 2.0 fluorometer (Thermo Fisher), while quality control and integrity measurement was performed using the TapeStation 4200 (Agilent).

Reverse transcription and template switching for library preparation was performed on total RNA samples using SMARTScribe Reverse Transcriptase, in line with the protocol specified in *Turchaninova et al., 2016*; Appendix 1—note 7. The reaction product was purified using SeraSure SPRI beads

(Appendix 1—note 7), then underwent three successive rounds of PCR, each of which was followed by a further round of bead purification. The first of these PCR reactions added a second strand to the reverse-transcribed cDNA and amplified the resulting DNA molecules; the second added partial Illumina sequencing adapters and further amplified the library, and the third added complete Illumina adapters, including i5 and i7 indices.

The concentration of each library was then quantified and the libraries were pooled in equimolar ratio, concentrated using SeraSure beads, and size-selected with the BluePippin (Sage Science) to obtain a purified amplicon band. Finally, following a final round of quality control, the pooled and size-selected libraries were sequenced on an Illumina MiSeq System (MiSeq Reagent Kit v3, 2 × 300 bp reads, 30% PhiX spike-in), either at the Cologne Center for Genomics (whole-body libraries) or with Admera Health (intestinal libraries).

### Data processing and analysis of repertoire data

Pre-processing of raw sequencing data (including quality filtering, consensus-read generation, and clonotyping) was performed using the pRESTO (*Vander Heiden et al., 2014*) and Change-O (*Gupta et al., 2015*) suites of command-line tools (Appendix 1—note 7, *Appendix 1—figure 1*). Downstream analysis of processed data, including diversity-spectrum inference (Appendix 1—note 7), RDI computation, GLM fitting and rarefaction, was performed in R, as was figure generation and all statistical tests. Generative model inference was performed using IGoR (*Marcou et al., 2018*). Snakemake (*Köster and Rahmann, 2012*) was used to design and run data-processing pipelines.

### Functional analysis of RNA-seq data

Intestinal RNA-seq data for gut cohort killifish (*Smith et al., 2017*) were obtained from SRA (BioProject accession PRJNA379208, *Supplementary file 2d*). Reads were mapped to the turquoise-killifish genome (*Reichwald et al., 2015*) with STAR (*Dobin et al., 2013*), using standard parameters, to compute raw read counts for each transcript and each individual. Read counts were normalized using DESeq2's default median-of-ratios method (*Love et al., 2014*). DESeq2 was then used to carry out differential expression analysis based on a generalized linear model, predicting abundance of each transcript in each individual given that individual's age and repertoire diversity (as calculated above). This analysis was repeated for each of six diversity orders (0, 1, 1.5, 2, 3, and 4).

Killifish transcripts were mapped to human orthologues with BioMart (*Durinck et al., 2009*; *Durinck et al., 2005*). In cases where multiple killifish transcripts mapped to a single human transcript, the individual fold change estimates produced by DESeq2 for the killifish transcripts were replaced by a single mean value. Transcripts were then ranked by fold change in descending order, and this ranked list was used as input for GSEA using ClusterProfiler's gseaGO function (*Wu et al., 2021*; *Yu et al., 2012*; *Subramanian et al., 2005*; *Mootha et al., 2003*), using the Benjamini-Hochberg method (*Benjamini and Hochberg, 1995*) to adjust for multiple comparisons and with a significance threshold of 0.05. This produced a list of GO terms (*Gene Ontology Consortium, 2021*; *Ashburner et al., 2000*) significantly enriched with respect to repertoire diversity, controlling for age. Redundant GO terms were summarized using ClusterProfiler's simplify function, with a similarity cutoff of 0.7.

Immune-associated GO terms were identified by descent from one of a small set of high-level immune-associated terms (*Supplementary file 3b*), which were identified manually. Terms descended from these manually selected ancestor terms were identified using the GO function GOBPOFFSPRING (*Gene Ontology Consortium, 2021*; *Ashburner et al., 2000*); any such descendant term was designated as immune-associated.

### Data and code availability

Raw data used in these analyses is available via NCBI (BioProject accession PRJNA662612). Processed data and code are available at https://github.com/willbradshaw/killifish-igseq/, (copy archived at swh:1:rev:2c933de6564c1055cb363389778f86bfa3fe4ab2; *Bradshaw, 2022*).

### Acknowledgements

We would like to thank Alexander Dilthey for detailed input on statistical approaches; Aleksandra Walczak, Thierry Mora, John Beausang, and Susana Magadan for their generous advice on designing the experimental protocols and analysis; and Jason Vander Heiden and Quentin Marcou for their

technical assistance with the pRESTO/Change-O and IGoR software. We would further like to thank all the members of the Valenzano lab for continuing support and feedback on the project; in particular, Joanna Dodzian and Patrick Smith for contributing experimental samples, and Davina Patel, Lena Schlautmann, and Linda Zirden for help with sample processing. This project was funded by the Max Planck Society, the Max Planck Institute for Biology of Aging, the CECAD Research Center in Cologne, and the DFG Collaborative Research Center 1310.

## Additional information

### Competing interests

Dario Riccardo Valenzano: Reviewing editor, eLife. The other authors declare that no competing interests exist.

### Funding

| Funder | Grant reference number | Author |
| --- | --- | --- |
| Max Planck Society | Valenzano lab budget | Dario Riccardo Valenzano |
| Deutsche Forschungsgemeinschaft | CRC 1310 | Dario Riccardo Valenzano |

The funders had no role in study design, data collection and interpretation, or the decision to submit the work for publication.

### Author contributions

William John Bradshaw, Data curation, Formal analysis, Investigation, Methodology, Software, Validation, Visualization, Writing – original draft, Writing – review and editing; Michael Poeschla, Formal analysis, Investigation; Aleksandra Placzek, Investigation; Samuel Kean, Formal analysis, Investigation, Software; Dario Riccardo Valenzano, Conceptualization, Project administration, Writing – original draft, Writing – review and editing

### Author ORCIDs

Dario Riccardo Valenzano (iD) http://orcid.org/0000-0002-8761-8289

### Decision letter and Author response

Decision letter https://doi.org/10.7554/eLife.65117.sa1
Author response https://doi.org/10.7554/eLife.65117.sa2

## Additional files

### Supplementary files

• Supplementary file 1. Software versions used in computational analyses.

• Supplementary file 2. Summary of killifish used in the study. (a) Summary of killifish used in IgSeq pilot and aging experiments. All fish are GRZ-AD strain and male, and hatched on May 9, 2016. (b) Summary of killifish used in IgSeq gut-microbiota transfer experiment. All fish are GRZ-Bellemans strain and male. (c) IDs and death weights of all individual turquoise killifish used in IgSeq aging experiment (a). (d) Individual killifish samples used in IgSeq gut-microbiota transfer (b).

• Supplementary file 3. Gene Ontology analysis. (a) Gene ontology (GO) terms identified by gene set enrichment analysis as being significantly associated with killifish repertoire diversity, controlling for age, in at least four different diversity orders. Ranks are determined based on absolute normalized enrichment value (NES), and reported both for all significantly enriched GO terms and for positively enriched terms only. Reported p-values have been adjusted for multiple testing using the Benjamini-Hochberg method (*Benjamini and Hochberg, 1995*) (Materials and methods). NES, rank, and p-value are summarized across all diversity orders for which a given GO term is significantly enriched, reported as mean ± standard deviation. (b) Parent terms used to identify immune-associated GO terms. Any term that is a descendant of any of these terms was identified as an immune term.

• Supplementary file 4. PCR conditions. (a) Master-mix components for reverse transcription reaction used in library preparation protocol (per sample). (b) Overall PCR thermocycler protocol for library preparation protocol (see (c) for missing parameter values). (c) Specific PCR protocols used for different stages of *Nothobranchius furzeri* immunoglobulin sequencing library preparation protocol (see (d) for overall thermocycler protocol and Supplemental (d) for cycle numbers). See *Supplementary file 5a* for sequence information. If the number of samples to be sequenced was small, all volumes of PCR three were doubled for a 50 µl total PCR volume. The stated volumes apply separately to both forward and reverse primers for each reaction. All primers were diluted to and stored at an initial concentration of 10 µM. See Supplemental (d) for specific cycle numbers used in each experiment. (d) PCR cycle numbers during *N. furzeri* IgSeq library preparation protocol (b–c). (e) Bead cleanups during *N. furzeri* IgSeq library preparation protocol. Each bead cleanup takes place immediately *after* its corresponding stage. Bead volumes are usually given as multiples of the sample volume. All elutions performed in the specified volume of elution buffer (10 mM Tris-HCl, pH 8.5). If the PCR three reaction volume differed from 25 µl, bead and elution volumes were rescaled proportionally to sample volume as appropriate. In each experiment, samples were pooled in equimolar ratio, so the input volume depended on the number of samples and the concentration of the libraries.

• Supplementary file 5. Primers and oligonucleotides used across the study. (a) Primers and oligonucleotides used in *Nothobranchius furzeri* IgSeq library preparation protocol. See *Appendix 1—figure 2* for more information about the structure of the template-switch adapter. For each P2 oligo (D701-D712), the N characters are replaced by the appropriate Illumina i7 index sequence from (c). For each P1 oligo (D501-D508), the highlighted N characters are replaced by the appropriate Illumina i5 index sequence from (d). The template-switch adapter and primer sequences homologous to it (M1SS and M1S) were provided by *Turchaninova et al., 2016*, while those homologous to constant-region exons (GSP, IGH-B, and IGH-C) were designed using Primer3 (*Untergasser et al., 2012*). (b) i5 index sequences. (c) i7 index sequences.

• Transparent reporting form

## Data availability

All data generated or analysed during this study are included in the manuscript and supporting files. Source data files have been provided. The raw data used in these analyses are available via NCBI (BioProject accession PRJNA662612). Processed data and code are freely available at https://github.com/will-bradshaw/killifish-igseq/ (copy archived at swh:1:rev:2c933de6564c1055cb363389778f86bfa3fe4ab2).

The following dataset was generated:

| Author(s) | Year | Dataset title | Dataset URL | Database and Identifier |
|---|---|---|---|---|
| Valenzano et al | 2020 | Antibody repertoire sequencing reveals systemic and mucosal immunosenescence in the short-lived turquoise killifish | http://www.ncbi.nlm.nih.gov/bioproject/?term=PRJNA662612 | NCBI BioProject, PRJNA662612 |

The following previously published dataset was used:

| Author(s) | Year | Dataset title | Dataset URL | Database and Identifier |
|---|---|---|---|---|
| Bradshaw W | 2020 | Antibody repertoire sequencing reveals systemic and mucosal immunosenescence in the short-lived turquoise killifish | https://github.com/willbradshaw/killifish-igseq | GitHub, killifish-igseq |

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

## Appendix 1

### Note 1: Hill diversity spectra

The Hill number or 'true diversity' $D$ of a population $X$ describes the *effective species richness* of that population: the number of equally common species that would be required to produce a population with the same overall diversity as $X$ (*Jost, 2006*). This value will increase with the number of species in the population, as well as with the evenness with which these species are distributed.

The term 'species' here can refer to any mutually exclusive and exhaustive set of classes into which elements in the population can be assigned; the nature of these classes will depend on the nature of the populations and elements being investigated. In the context of immune-repertoire sequencing, a 'population' is an individual repertoire, an 'element' is a unique sequence, and a 'species' will typically refer to either clones (sequences descended from a single naive ancestor B-cell) or V(D)J gene-segment combinations. In this paper, Hill diversities computed using clones as species are referred to as *clonal diversities*, while those computed using VJ segment combinations as species are referred to as *VJ diversities*.

In addition to the species composition of, $X$ the Hill number depends on the diversity order, $q$ which denotes the degree to which less-abundant species are downweighted relative to more-abundant species when calculating diversity. At a given value of $q$, the Hill number of $X$ is given by:

$$^qD(X) = \left(\sum_{s \in S} p_s^q\right)^{\frac{1}{1-q}}$$

(1)

where $S$ is the set of species in $X$ and $p_s$ is the relative frequency of a species $s$ (i.e. the fraction of all elements that belong to $s$). Note that, as higher values of $q$ progressively downweight the contributions of less-abundant species, $^qD(X)$ is monotonically decreasing with increasing $q$. As each diversity order captures a different aspect of the diversity structure of the population, plotting the value of $^qD(X)$ against $q$ – a diversity *spectrum* – gives a more informative view of the diversity of $X$ than any single diversity order alone.

Importantly, many other common diversity metrics can be converted into Hill numbers through simple transformations: $^0D(X)$ gives species richness, $^2D(X)$ gives the reciprocal of Simpson's index, $q \xrightarrow{lim} \infty \, ^qD(X)$ gives the reciprocal of the Berger-Parker index, and $q \xrightarrow{lim} \infty \, ^qD(X)$ gives the exponential of the Shannon entropy (*Jost, 2006*; *Miho et al., 2018*). Hill diversity measurements of the appropriate order can therefore be easily converted to and from these other diversity metrics if needed. In practice, the exponential Shannon entropy is typically substituted for $^1D(X)$ (which is undefined) to produce a continuous function when plotting Hill diversity spectra:

$$^qD(X) = \begin{cases} \left(\sum_{s \in S} p_s^q\right)^{\frac{1}{1-q}} & q \neq 1 \\ \exp\left(\sum_{s \in S} p_s \cdot \ln p_s\right) & q = 1 \end{cases}$$

(2)

### Note 2: Alpha and beta diversity

In some cases, a population $C$ can be partitioned into some number of disjoint *subpopulations* $X_1$, $X_2, \ldots, X_M$, each with its own set of species $S_X$ and species frequencies $p_X, s$. The diversity of $C$ can then be conceptualized in several ways (*Jost, 2007*):

The **gamma diversity** of $C$ is the diversity across the whole population, ignoring its subpopulation structure.

The **alpha diversity** of $C$ is the diversity arising from differences in species identity among individuals *within* each subpopulation, and is given by a weighted generalized average of the diversities of those subpopulations; it can be thought of as the expected diversity of a single subpopulation drawn at random from $C$.

The **beta diversity** of $C$ is the diversity arising from variability in species composition *between* subpopulations; it is lowest when all subpopulations have identical species composition and highest when they have no species in common.

In terms of Hill diversity, $^qD_\gamma(C)$ gives the effective richness of the whole population, ignoring its subpopulation structure; $^qD_\alpha(C)$ gives the effective richness of an 'average'' subpopulation from $C$, and $^qD_\beta(C)$ gives the effective number of disjoint subpopulations in $C$ (i.e. the number of subpopulations with no species in common that would give rise to the same gamma diversity, given an alpha-diversity value).

The alpha and beta diversities of a population are independent; two different structured populations can have identical alpha and very different beta diversities, or vice versa, depending on the exact species compositions of their subpopulations. The alpha and beta diversity of a structured population also completely determine its gamma diversity (*Jost, 2007*):

$$D_\gamma(C) = D_\alpha(C) \times D_\beta(C) \tag{3}$$

and therefore

$$D_\beta(C) = \frac{D_\gamma(C)}{D_\alpha(C)} \tag{4}$$

This (*Equation 4*) is typically the easiest way of computing the beta diversity of a given structured population. Note that, as the quotient of two monotonically decreasing functions, $D_\beta(C)$ is itself not necessarily monotonically decreasing.

Under this framework, the diversities of order $q$ for a structured population $C$ are given (*Jost, 2007*) by:

$$^qD_\gamma(C) = \left[ \frac{\sum\limits_{s \in S} \left( \sum\limits_{X \in C} w_X p_{X,s} \right)}{\left( \sum\limits_{X \in C} w_X \right)} \right]^{\frac{1}{1-q}} \tag{5}$$

$$^qD_\alpha(C) = \left[ \frac{\sum\limits_{X \in C} \sum\limits_{s \in S_X} (w_X p_{X,s})^q}{\left( \sum\limits_{X \in C} w_X \right)} \right]^{\frac{1}{1-q}} \tag{6}$$

$$^qD_\beta(C) = \frac{^qD_\gamma(C)}{^qD_\alpha(C)} \tag{7}$$

where   is the relative weighting of a subpopulation $X$. When all the subpopulations have equal sizes $N_X = \frac{N}{M}$ and equal weights, $w = \frac{1}{M}$ as is the case with our repertoire data, these formulae simplify to:

$$^qD_\gamma(C) = \left[ \frac{\sum\limits_{s \in S} \left( \sum\limits_{X \in C} p_{X,s} \right)^q}{M^q} \right]^{\frac{1}{1-q}} \tag{8}$$

$$^qD_\alpha(C) = \left[ \frac{\sum\limits_{X \in C} \sum\limits_{s \in S_X} (p_{X,s})^q}{M} \right]^{\frac{1}{1-q}} \tag{9}$$

These equations are valid for all values of $q \in \mathbb{R}$ except 1, providing a spectrum of alpha-, beta-, or gamma-diversity measures analogous to the diversity spectra provided for simple populations in Appendix 1—note 1. As in that case, a special case needs to be made for $q=1$ in order to make these functions continuous, substituting the limit of the diversity as $q \to 1$ for the undefined value at $q=1$

$$^1D_\gamma(C) = \exp \left( -\sum\limits_{s \in S} p_s \cdot \ln p_s = \right) {}^1D(C) \tag{10}$$

$$^1D_\gamma(C) = \exp\left[\frac{1}{M}\sum_{X\in C}\left(-\sum_{s\in S_X}p_{X,s}\cdot\ln p_{X,s}\right)\right] = \exp\left[\frac{1}{M}\sum_{X\in C}\ln{}^1D(X)\right] \tag{11}$$

More detailed derivations of *Equation 8* to *Equation 11* are available upon request.

## Note 3: Rescaling beta diversity

As discussed in Appendix 1—note 2, while alpha and gamma diversity are expressed in terms of an effective number of species, beta diversity is expressed in terms of an effective number of subpopulations. Since the effective number of subpopulations is determined in part by the actual number of subpopulations, this means that the beta diversity, unlike alpha and gamma diversity, is directly dependent on the number of subpopulations $M$. If two different structured populations contain different numbers of subpopulations, it is therefore not possible to compare their beta diversity values directly; first, the beta diversity spectra of the populations must be *rescaled* to a common range.

The *minimum* beta diversity of a structured population obtains when all subpopulations have identical species composition. In this case, the beta diversity for the structured population is equal to 1. The *maximum* beta diversity, meanwhile, obtains when there is no overlap in species between any pair of subpopulations in the structured population. In this case beta diversity for the structured population is equal to the number of subpopulations $M$ (derivations available upon request).

The beta diversity for a structured population with $M$ subpopulations therefore ranges between 1 (identical composition) and $M$ (maximally divergent composition). The beta diversities of such a population can thus be transformed onto a new scale from 0 (minimum beta diversity) to 1 (maximum beta diversity) as follows:

$$^qD_{\beta\,\text{rescales}}(C) = \frac{^qD_\beta(C) - {}^qD_{\beta\,\min}(C)}{^qD_{\beta\,\max}(C) - {}^qD_{\beta\,\min}(C)} = \frac{^qD_\beta(C) - 1}{M - 1}$$

By transforming the beta-diversity spectra of different structured populations onto this common scale, the inter-subpopulation variability of those populations can be meaningfully compared, even if they differ in the number of subpopulations they contain.

## Note 4: IgSeq library preparation

To reverse-transcribe antibody heavy-chain transcripts from total RNA samples, and to attach template-switch adapter oligos and UMIs, 750 ng of total RNA from a killifish sample was combined with 2 µl of 10 µmol gene-specific primer (GSP, *Supplementary file 5a*), homologous with the second constant-region exon of *N. furzeri* IgM ($C_\mu^2$) (*Bradshaw and Valenzano, 2020*). The reaction volume was brought to a total of 8 µl with nuclease-free water, and the resulting mixture was incubated for 2 min at 70°C to denature the RNA, then cooled to 42°C to anneal the GSP (*Turchaninova et al., 2016*). Following annealing, the RNA-primer mixture was combined with 12 µl of reverse transcription master-mix (*Supplementary file 4a*), including the reverse-transcriptase enzyme and template-switch adapter SmartNNNa, *Supplementary file 5a* and *Appendix 1—figure 2*, sequence provided in *Turchaninova et al., 2016*. The complete reaction mixture was incubated for 1 hr at 42°C for the reverse transcription reaction, then mixed with 1 µl of uracil DNA glycosylase (QIAGEN, 100 mg/ml) and incubated for a further 40 min at 37°C to digest the template-switch adapter.

Following reverse transcription, the reaction product was purified using SeraSure SPRI (solid-phase reversible immobilization) bead preparation (*Fisher et al., 2011*) (Appendix 1—note 5, *Supplementary file 4e*). The reaction mixture then underwent three successive rounds of PCR with 2× Kapa HiFi HotStart ReadyMix PCR Kit (Kapa Biosystems), using the general reaction protocol described in *Supplementary file 4b* and reaction-specific volumes and parameters from *Supplementary file 4c*. Each round of PCR was followed by a further round of bead cleanups (Appendix 1—note 5, *Supplementary file 4e*) to purify the reaction product prior to the next reaction.

Following the final PCR and cleanup, the total concentration of each library was assayed with the Qubit 2.0, while the size distribution of each library was obtained using the TapeStation 4200. To obtain the concentration of complete library molecules in each case (as opposed to primer dimers or other off-target bands), the ratio between the concentration of the desired library band (c. 620–680 bp) and the total concentration of the sample was calculated for each TapeStation lane, and the total concentration of each library as measured by the Qubit was multiplied by this ratio to obtain

an estimate of the desired quantity. All the libraries for a given experiment were then pooled, such that the estimated concentration of each library in the final pooled sample was equal and the total mass of nucleic acid in the pooled sample was at least 240 ng. The pooled libraries underwent a final bead cleanup (*Supplementary file 4e*) to concentrate the samples, and the beads were separated and removed from the sample.

The pooled and concentrated samples underwent size selection with the BluePippin (Sage Science, 1.5% DF Marker R2, broad 400–800 bp target band) according to the manufacturer's instructions. The size-selected libraries then underwent a final round of quality control with the Qubit and TapeStation (as above) to confirm their collective concentration (at least 1.5 nmol) and size distribution. Finally, the pooled and size-selected libraries were sequenced on an Illumina MiSeq System (MiSeq Reagent Kit v3, 2 × 300 bp reads, 30% PhiX spike-in) as described in the main text.

## Note 5: Nucleic acid cleanup with SPRI beads

To prepare 50 ml of SeraSure bead suspension, a stock of SeraMag beads (GE Healthcare, 50 mg/ml) was vortexed thoroughly, and 1 ml was transferred to a new tube. This tube was then transferred to a magnetic rack and incubated at room temperature for 1 min, then the supernatant was removed and replaced with 1 ml TET buffer (10 mM Tris base, 1 mM $Na_2$-EDTA, 0.05% (v/v) Tween 20, pH 8.0) and the tube was removed from the rack and vortexed thoroughly. This washing process was repeated twice more, for a total of three washes in TET. A fourth cycle was used to replace the TET with incomplete SeraBind buffer (4.2 M NaCl, 16.8 mM Tris base, 1.68 mM $Na_2$-EDTA, pH 8.0). The vortexed 1 ml aliquot of beads was then transferred to a conical tube containing 28 ml incomplete SeraBind buffer and mixed by inversion. Twenty ml 50% (w/v) PEG 8000 solution was dispensed slowly down the side of the conical tube, bringing the total volume to 49 ml. Finally, this was brought to 50 ml by adding 250 μl 10% (w/v) Tween 20 solution and 750 μl autoclaved water to complete the SeraSure bead suspension.

To perform a bead cleanup, an aliquot of prepared SeraSure suspension was vortexed thoroughly to completely resuspend the beads, then the appropriate relative volume of SeraSure suspension was added to a sample, mixing thoroughly by gentle pipetting. The sample was incubated at room temperature for 5 min to allow the beads to bind the DNA, then transferred to a magnetic rack and incubated for a further 5 min to draw as many beads as possible out of suspension. The supernatant was removed and discarded and replaced with 80% ethanol, to a volume sufficient to completely submerge the bead pellet. The sample was incubated for 0.5–1 min, then the ethanol was replaced and incubated for a further 0.5–1 min. The second ethanol wash was removed, and the tube left on the rack until the bead pellet was almost, but not completely, dry, after which it was removed from the rack. The bead pellet was resuspended in a suitable volume of elution buffer (10 mM Tris-HCl, pH 8.5) then incubated at room temperature for at least 5 min to allow the nucleic acid molecules to elute from the beads.

Unless otherwise specified, the beads from a cleanup were left in a sample during subsequent applications. To remove beads from a sample, the sample was mixed gently but thoroughly to resuspend the beads, incubated for an extended time period (at least 10 min) to maximize nucleic acid elution, then transferred to a magnetic rack and incubated for 2–5 min to remove the beads from suspension. The supernatant (containing the eluted nucleic acid molecules) was then transferred to a new tube, and the beads discarded.

## Note 6: Pre-processing repertoire-sequencing data

Unless otherwise specified, utilities used in the pre-processing pipelines were provided by the pRESTO (*Vander Heiden et al., 2014*) and Change-O (*Gupta et al., 2015*) suites of command-line tools. All code used as part of this pipeline, as well as for downstream analysis and visualization, will be made available at https://github.com/willbradshaw/killifish-igseq/, (copy archieved at swh:1:rev:2c933de6564c1055cb363389778f86bfa3fe4ab2; *Bradshaw, 2022*).

Demultiplexed, adapter-trimmed MiSeq sequencing data were uploaded by the sequencing provider to Illumina BaseSpace and accessed via the Illumina utility program BaseMount. Library annotation information (fish ID, sex, strain, age at death, death weight, etc.) was added to the read headers of each library FASTQ file, and library was assigned a replicate and individual identity. Reads from different sequencing libraries were pooled together, then split by replicate identity; this pooling and re-splitting process enables all reads considered to be a single replicate to be processed

together even if sequenced separately, maximizing the effectiveness of UMI-based pre-processing, while also allowing all replicates to be processed in parallel.

After pooling and re-splitting, the raw read set underwent quality control, discarding any read with an average Phred score of less than 20. Invariant primer sequences were removed and UMI sequences identified and extracted.

In order to reduce the level of barcode errors in each dataset, primer-masked reads then underwent barcode clustering, in which reads with the same replicate identity and highly similar UMI sequences were grouped together into the same molecular identifier group (MIG). To do this, 5'-reads were first clustered by UMI sequence using a 90% sequence identity cutoff. In order to split any genuinely distinct MIGs accidentally united by this process, as well as to reduce the level of barcode collisions, the reads within each MIG were clustered again based on their insert sequences; the cluster dendrogram was cut at 75% total sequence identity, and each subcluster was separated into its own distinct MIG. These clustering thresholds (90% for barcode clustering, 75% for barcode splitting) were identified empirically as the values that maximize the number of reads passing downstream quality checks in turquoise-killifish data.

The cluster annotations from these two clustering steps were combined into a single annotation, uniquely identifying each MIG in each replicate. These annotations were further modified to designate the replicate identity of each read, giving each MIG a unique annotation across the entire dataset. These annotations were copied to the reverse reads, such that each read pair had a matching MIG annotation, and reads without a mate (due to differential processing of the two reads files) were discarded. The 5'-reads were then grouped based on MIG identity, and the reads in each cluster grouping were aligned and collapsed into a consensus read sequence; an identical consensus-read-generation step was performed on the reverse reads.

After consensus-read generation had been performed for both 5'- and 3'-reads, the annotations attached to each read were again unified across read pairs with matching cluster identities, and consensus reads without a mate of the same cluster identity were dropped. Consensus-read pairs with matching cluster annotations were then aligned and merged into a single contiguous sequence. To convert this dataset of MIGs (representing distinct RNA molecules) into one of unique sequences (representing distinct B-cells), merged consensus sequences with identical insert sequences but distinct cluster identities were collapsed together into a single FASTQ entry, recording the number, size, and UMI makeup of contributing MIGs in each case. Sequences represented by only a single read across all MIGs in the dataset were discarded as unreliable, and the remaining non-singleton sequences from all replicates were combined into a single FASTA file.

V/D/J identities were assigned to each sequence using IgBLAST (*Ye et al., 2013*), and the sequences imported into a tab-delimited Change-O database file for downstream processing. Clonotype identities inferred using Change-O's nearest-neighbor distance-threshold technique (*Gupta et al., 2015*; *Nouri and Kleinstein, 2018*), with replicates from the same individual pooled before undergoing clonotyping to enable comparison of clonotype composition across replicates. A full-length germline sequence was constructed for each sequence entry, and improved V/D/J assignments were made based on the assignments of other sequences in the same clone. Finally, each sequence in the dataset was annotated according to whether or not it possessed V/D/J assignments and whether these assignments were ambiguous (i.e. whether multiple possible assignments were given rather than just one), and combined VJ and VDJ assignments were obtained by concatenating the individual segment assignments as appropriate. The processed sequence databases were then passed on to downstream analysis pipelines.

## Note 7: Diversity-spectrum inference and statistical comparison

Methods for computing diversity spectra were adapted from those presented in the Alakazam R package (*Gupta et al., 2015*; *Stern et al., 2014*), following completion of the pre-processing pipeline described in Appendix 1—note 6.

To ensure comparability between individuals, and obtain estimates of confidence intervals for diversity estimates, the repertoire of each individual underwent multinomial bootstrap resampling: for each of 2000 bootstrap replicates, the same number of unique sequences was sampled, with replacement, from each individual. Following bootstrapping, Hill diversity estimates for each bootstrap replicate and diversity order were calculated as appropriate for the type of diversity being estimated:

- **Individual diversity** spectra (i.e. for individual fish, not aggregated by age group) were computed as specified in *Equation 2*.
- **Alpha-diversity** spectra for each age group were computed across all individuals in that age group as specified in *Equation 9* and *Equation 11*.
- **Gamma-diversity** spectra for each age group were computed across all individuals in each age group as specified in *Equation 8* and *Equation 10*.
- **Beta-diversity** spectra for each age group were computed as specified in *Equation 7*, and rescaled as described in Appendix 1—note 3.

Having computed diversity values for each bootstrap replicate, diversity spectra were visualized as the mean across all replicates, while 95% confidence intervals were computed empirically as the 0.025- and 0.975-quantiles of diversity across replicates.

To estimate the statistical significance of an age effect on individual repertoire diversity, we adopted a permutation-based approach. For each of 3000 permutations, the age-group assignments of the individuals in the dataset were randomly reshuffled, and a Kruskal-Wallis one-way analysis-of-variance test for an age effect was performed on the individual diversity estimates for each permutation, bootstrap replicate, and diversity order. For each permutation and diversity order, the largest Kruskal-Wallis $H$ statistic across all bootstrap replicates was identified. The same process was repeated on the (unpermuted) real data for each bootstrap replicate and diversity order, again taking the largest $H$ statistic across all bootstrap replicates for each diversity order. The p-value of the age effect at a given diversity order was then computed as the proportion of permutations with a higher maximum $H$ value than that obtained from the real data.

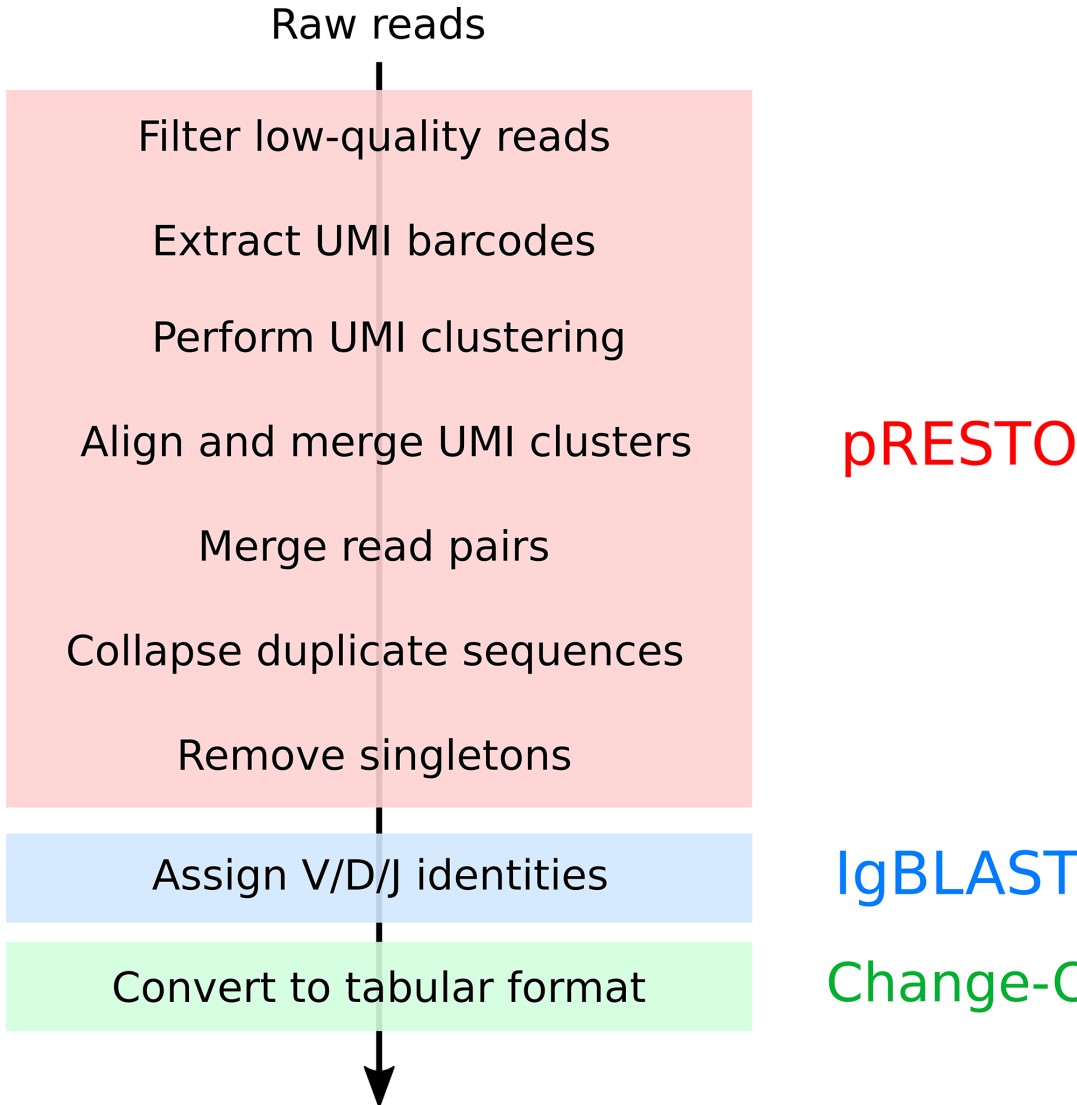

**Appendix 1—figure 1.** Summary of pre-processing pipeline applied to killifish repertoire-sequencing data (Appendix 1—note 6).

AAGCAGUGGTAUCAACGCAGAGUNNNN—

—UNNNNUNNNNUCTTrGrGrGrG

**Appendix 1—figure 2.** The SmartNNNa template-switch adapter. Annotated sequence of the SmartNNNa barcoded template-switch adapter (TSA) used in template-switch reverse transcription for immunoglobulin-sequencing library preparation (*Turchaninova et al., 2016*). The 5'-terminal pink characters represent an invariant sequence used for primer-binding in downstream PCR steps, while the gray N characters represent the random nucleotides constituting the unique molecular identifier (UMI), each of which could take any value from A, C, G, or T. The blue U residues represent deoxyuridine, which is specifically digested after reverse transcription to remove residual TSA oligos from the reaction mixture. The orange, 3'-terminal rG characters indicate riboguanosine residues, which pair with terminal-transferase added cytidine residues to enable template switching.

