## [Editor Report]

This study introduces the killifish as a short-lived vertebrate model for immune aging and immunosenescence and characterizes the changes in the immune-repertoire during aging. This work provides an important first step in understanding how aging impacts the immune system in this model organism and will set the stage for many future studies.

---

## [Decision Letter]

**Decision letter after peer review:**

Thank you for submitting your article "Antibody repertoire sequencing reveals systemic and mucosal immunosenescence in the short-lived turquoise killifish" for consideration by *eLife*. Your article has been reviewed by 2 peer reviewers, including Matt Kaeberlein as the Reviewing Editor and Reviewer #2, and the evaluation has been overseen by a Senior Editor.

Essential revisions:

This is a very interesting study introducing the killifish as a potential model for immune aging and immunosenescence and characterizing the changes in age-associated immune-repertoire. The authors convincingly show a decrease in diversity of the large expanded B-cell clones that is greater than small clones and a more pronounced change in the intestinal antibody repertoire with age. A limitation of the current study is its descriptive nature and lack of mechanistic insight or strong evidence that aging killifish truly experience functional immunosenescence. We have a few suggestions to potentially enhance the impact of this work that we would like the authors to consider and respond to:

1. Addition of functional data showing a decline in adaptive immunity that goes along with the loss of diversity in the antibody repertoire or citation and discussion of prior literature supporting this in killifish. As it is, it is difficult to know the extent to which the observed changes are strongly correlated with changes in immune function.

2. Whole genome sequencing of lymphoid tissues and brain as a control, from the same old fish to determine whether there are clonal somatic mutations. If confirmed, it may be an important finding, as it would mean that clonal expansions emerge as fast as the killifish lifespan, and it would be a great model to study mechanisms of mutation accumulation and clonal selection with age. This WGS data may be further used to reconstruct immunoglobulin repertoires to understand if the whole-body decrease is driven solely by intestine B cells, or it initiates in lymphoid tissues.

3. RNA sequencing of intestine samples or spleen from young versus old killifish to obtain insights into possible molecular mechanisms clonal expansion and diversity loss. Spleen RNA sequencing may be used to reconstruct the immunoglobulin repertoire. The authors used 750 ng of total RNA in the current study, so there should be enough material for RNA sequencing. Or single cell RNA sequencing may be performed.

While the lack of functional or mechanistic data does not necessarily preclude publication in *eLife*, the manuscript as is currently overstates the demonstrated importance of this phenomenon. At a minimum, it should be be explicitly noted that further research is needed to determine whether this actually represents immune senescence or simply changes that are of unknown consequence.

*Reviewer #2 (Recommendations for the authors):*

This is a very interesting study introducing the killifish as a potential model for immune aging and immunosenescence and characterizing the changes in age-associated immune-repertoire. The authors convincingly show a decrease in diversity of the large expanded B-cell clones that is greater than small clones and a more pronounced change in the intestinal antibody repertoire with age. The biggest weakness with the current study is its descriptive nature and lack of strong evidence that these animals truly experience functional immunosenescence. The impact of this work could be strengthened by functional data showing a decline in adaptive immunity that goes along with the loss of diversity in the antibody repertoire or citation and discussion of prior literature supporting this in killifish. As it is, it is difficult to know the extent to which the observed changes are strongly correlated with changes in immune function. While the lack of functional data does not necessarily preclude publication in *eLife*, the manuscript as is currently overstates the demonstrated importance of this phenomenon. It should at least be explicitly noted that further research is needed to determine whether this actually represents immune senescence or simply changes that are of unknown consequence.

---

## [Author Response]

Essential revisions:This is a very interesting study introducing the killifish as a potential model for immune aging and immunosenescence and characterizing the changes in age-associated immune-repertoire. The authors convincingly show a decrease in diversity of the large expanded B-cell clones that is greater than small clones and a more pronounced change in the intestinal antibody repertoire with age. A limitation of the current study is its descriptive nature and lack of mechanistic insight or strong evidence that aging killifish truly experience functional immunosenescence. We have a few suggestions to potentially enhance the impact of this work that we would like the authors to consider and respond to:1. Addition of functional data showing a decline in adaptive immunity that goes along with the loss of diversity in the antibody repertoire or citation and discussion of prior literature supporting this in killifish. As it is, it is difficult to know the extent to which the observed changes are strongly correlated with changes in immune function.

These points are well taken. While our initial submission demonstrates that the diversity of the killifish repertoire declines with age, it is true that this does not necessarily imply that this decline is linked to changes in immune functionality.

To provide functional insights into the transcriptomic signature associated with different antibody diversity orders, we now include an analysis linking repertoire diversity data in our intestinal cohort to pre-existing intestinal RNA-seq data from the same individuals (Figure 6). The combination of these two data sets allows us to analyse changes in gene expression with respect to intestinal antibody diversity, controlling for age. We find that a number of immune-activity GO terms – including “B cell receptor signaling pathway”, “B cell proliferation”, and “lymphocyte activation” are significantly positively enriched with respect to repertoire diversity across multiple diversity orders. A decline in intestinal antibody diversity – as seen in ageing – is thus associated with a decline in B-cell immune activity in killifish.

We acknowledge that confident demonstration of a causal link between repertoire diversity and immune state will require experimental challenge of host immunity, for example through infection experiments – something we will address in the future and is beyond the scope of this work. However, we believe these new data are sufficient to demonstrate a significant association between the two, supporting the biological relevance of the age-associated decline in diversity we observe.

2. Whole genome sequencing of lymphoid tissues and brain as a control, from the same old fish to determine whether there are clonal somatic mutations. If confirmed, it may be an important finding, as it would mean that clonal expansions emerge as fast as the killifish lifespan, and it would be a great model to study mechanisms of mutation accumulation and clonal selection with age. This WGS data may be further used to reconstruct immunoglobulin repertoires to understand if the whole-body decrease is driven solely by intestine B cells, or it initiates in lymphoid tissues.

We agree that further investigation of primary repertoire development in killifish lymphoid organs would be a valuable direction for future work, and would help disentangle whole-body from intestine-specific repertoire changes. However, we believe our current analysis is sufficient to demonstrate the presence of clonal somatic mutations in the whole-body repertoire. The pRESTO/Change-O pipeline used in our analysis can distinguish heavy-chain sequences arising from different naive ancestors, and the presence of large clones in the killifish repertoire (see e.g. Supplemental Figure 5A) necessitates rapid clonal expansion.

Ongoing work in our group is indeed directed at studying somatic DNA sequence variation across tissues during aging in killifish, including alternative experimental approaches to investigating killifish repertoire aging. We have now added a sentence about these further research directions to the manuscript discussion. However, we feel these further experiments may be beyond the specific scope of the present work, which is focused on high-level changes in killifish antibody repertoire composition with age.

3. RNA sequencing of intestine samples or spleen from young versus old killifish to obtain insights into possible molecular mechanisms clonal expansion and diversity loss. Spleen RNA sequencing may be used to reconstruct the immunoglobulin repertoire. The authors used 750 ng of total RNA in the current study, so there should be enough material for RNA sequencing. Or single cell RNA sequencing may be performed.

We thank the reviewer for these suggestions. We certainly agree that investigation of repertoire aging in a wider array of immune organs, including spleen, would be highly valuable, and that killifish is a promising model organism in which to carry out these investigations. We have now included analysis of RNA-sequencing data from the killifish gut, which as discussed above supports an association between loss of repertoire diversity and immune function in that organ (see response to A.1). We hope for future work to more comprehensively explore the landscape of organ-specific repertoire ageing in the turquoise killifish; however, we feel that this would be beyond the scope of the present study.

While the lack of functional or mechanistic data does not necessarily preclude publication in eLife, the manuscript as is currently overstates the demonstrated importance of this phenomenon. At a minimum, it should be be explicitly noted that further research is needed to determine whether this actually represents immune senescence or simply changes that are of unknown consequence.

We hope our new analyses and the rewriting of the manuscript address these major points.

Reviewer #2 (Recommendations for the authors):This is a very interesting study introducing the killifish as a potential model for immune aging and immunosenescence and characterizing the changes in age-associated immune-repertoire. The authors convincingly show a decrease in diversity of the large expanded B-cell clones that is greater than small clones and a more pronounced change in the intestinal antibody repertoire with age.

We thank the reviewer for their supportive assessment of our work!

The biggest weakness with the current study is its descriptive nature and lack of strong evidence that these animals truly experience functional immunosenescence. The impact of this work could be strengthened by functional data showing a decline in adaptive immunity that goes along with the loss of diversity in the antibody repertoire or citation and discussion of prior literature supporting this in killifish. As it is, it is difficult to know the extent to which the observed changes are strongly correlated with changes in immune function. While the lack of functional data does not necessarily preclude publication in eLife, the manuscript as is currently overstates the demonstrated importance of this phenomenon. It should at least be explicitly noted that further research is needed to determine whether this actually represents immune senescence or simply changes that are of unknown consequence.

We agree that our original submission did not sufficiently address the question of functional relevance of an age-related decline in repertoire diversity. Previous work has broadly indicated an age-related immune decline in killifish, and we have now referenced and discussed that work more clearly in our manuscript. However, this is not sufficient to show that an age-related decline in repertoire diversity, as we observe, is *per se* linked to immune dysfunction.

To address this point, we now include an analysis linking repertoire diversity data in our intestinal cohort to pre-existing intestinal RNA-seq data from the same individuals. The combination of these two data sets allows us to analyse changes in gene expression with respect to intestinal antibody diversity, controlling for age. We find that a number of immune-activity GO terms – including “B cell receptor signaling pathway”, “B cell proliferation”, and “lymphocyte activation” are significantly positively enriched with respect to repertoire diversity across multiple diversity orders. A decline in intestinal antibody diversity – as seen in ageing – is thus associated with a decline in B-cell immune activity in that organ (Figure 6).

Further work in this area should investigate the relationship between diversity and immune function in other organs, as well as investigating whether these connections are causally linked to altered response to pathogens. We have edited the discussion to make these future research needs clear. However, we believe these results are sufficient to demonstrate a meaningful relationship between repertoire diversity and immune function.